# Influence of Nutrition on Growth and Development of Metabolic Syndrome in Children

**DOI:** 10.3390/nu16223801

**Published:** 2024-11-06

**Authors:** Alessia Quarta, Maria Teresa Quarta, Concetta Mastromauro, Francesco Chiarelli, Cosimo Giannini

**Affiliations:** Department of Pediatrics, University of Chieti—Pescara, G. D’Annunzio, 66100 Chieti, Italy; alessiaquarta54@gmail.com (A.Q.); terryquarta@gmail.com (M.T.Q.); tittimastromauro@gmail.com (C.M.); chiarelli@unich.it (F.C.)

**Keywords:** nutrition, growth, children, metabolic programming, obesity, metabolic syndrome

## Abstract

Obesity is currently an increasing public health burden due to its related metabolic and cardiovascular complications. In Western countries, a significant number of people are overweight or obese, and this trend is, unfortunately, becoming increasingly common even among the pediatric population. In this narrative review, we analyzed the role of nutrition during growth and its impact on the risk of developing metabolic syndrome and cardiovascular complications later in life. An impactful role in determining the phenotypic characteristics of the offspring is the parental diet carried out before conception. During intrauterine growth, the main risk factors are represented by an unbalanced maternal diet, excessive gestational weight gain, and impaired glycemic status. Breastfeeding, on the other hand, has many beneficial effects, but at the same time the quality of breast milk may be modified if maternal overweight or obesity subsists. Complementary feeding is likewise pivotal because an early introduction before 4 months of age and a high protein intake contribute to weight gain later. Knowledge of these mechanisms may allow early modification of risk factors by implementing targeted preventive strategies.

## 1. Introduction

Nutrition is a fundamental factor regulating growth from intrauterine life to childhood. Unbalanced nutritional status during intrauterine life can result in impaired fetal growth, known as intrauterine growth restriction (IUGR). In this condition, several dysfunctions arise as a consequence of fetal adaptation to malnutrition, particularly the development of insulin resistance (IR) and the reduction of insulin production [1]. The ability to store fat to survive in unfavorable metabolic conditions is the basis of the “thrifty phenotype” theory: malnutrition during intrauterine life causes epigenetic changes that promote energy intake. Indeed, metabolic adaptive alterations can affect glucose–insulin metabolism, leading to obesity, IR, and type 2 diabetes (T2D) later in life. Malnutrition is not the only condition associated with impaired intrauterine growth. Other factors include placental dysfunction, stress, hypoxia, prenatal glucocorticoids, xenobiotics such as endocrine disruptors, maternal smoking, and alcohol or drug intake during pregnancy [2]. Maternal obesity also poses a risk for the development of infant adiposity due to changes in human milk oligosaccharide (HMO) concentrations, negatively impacting HMO sialylation [3]. Nutrition plays a crucial role in defining body composition, especially during adolescence. According to a World Health Organization (WHO) report, one out of ten children and adolescents ages 5–17 years are obese, and this proportion is rapidly increasing worldwide [4]. Various definitions of metabolic syndrome in children have been proposed, but there is a lack of consensus on which to use. The International Diabetes Federation (IDF) proposed a pediatric metabolic syndrome definition based on the adult criteria, applying it only to children over 10 years old. Criteria for defining metabolic syndrome include five risk factors: hyperglycemia, hypertriglyceridemia, central adiposity, elevated blood pressure, and low high-density lipoprotein cholesterol (HDL-C) [5]. Some risk factors for metabolic syndrome occur during the prenatal and early postnatal periods, such as maternal gestational diabetes mellitus (GDM), low birth weight, improper infant feeding practices, and early adiposity rebound. Moreover, an increased incidence of cardiovascular comorbidities is observed in childhood as a consequence of obesity; particularly, high total cholesterol, high body mass index (BMI), and high low-density lipoprotein cholesterol (LDL-C) are correlated with increased carotid artery intima-media thickness, a marker of atherosclerosis [6]. A strong association between high BMI in childhood and coronary artery disease is described in the literature. This association is considered the cause of vascular damage that begins in early childhood and progresses to an end-stage process by mid-adulthood [7]. Indeed, obesity and IR cause oxidative stress and chronic inflammation, leading to endothelial dysfunction. C-reactive protein (CRP) and urinary prostaglandin F2 (PGF-2) levels can predict oxidative stress and chronic inflammation in high-risk obese children [8]. On the other hand, pro-inflammatory mediators such as tumor necrosis factor alpha (TNF-α), interleukin-6 (IL-6), and CRP promote lipolysis by activating macrophages. This causes ectopic lipid accumulation, promoting the onset of IR. Furthermore, IR, leptin excess, and hypoxia during obstructive sleep apnea (OSA), favored by obesity, activate the sympathetic nervous system, promoting the development of hypertension [9,10]. These conditions, which are part of the complex picture of metabolic syndrome, increase individual cardiovascular risk, leading to a higher incidence of morbidity and mortality in the adult population [11].

## 2. Materials and Methods

We conducted a narrative review of the literature to identify studies examining the effects of nutrition on the development of obesity and metabolic syndrome during childhood. We searched for relevant articles in the PubMed database using keywords and keyword combinations, such as “nutrition, fetal growth, metabolic programming, gestational diabetes mellitus, maternal diet, birthweight, infant feeding, weaning, obesity, metabolic syndrome, and children”. Inclusion criteria were recent studies on nutrition and metabolic health published since 2010 with an emphasis on systematic reviews, meta-analyses, observational studies, and randomized controlled trials (RCTs). Exclusion criteria included articles not in English and studies from sources deemed unreliable. After screening titles, abstracts, and full texts, we included 17 papers in this narrative review.

## 3. Growth and Regulating Factors: Nutrition and Hormones

Growth represents a complex developmental process leading to the acquisition of an individual’s definitive morphological and functional characteristics. Its regulation depends on multiple aspects, including genetic, epigenetic, environmental, and hormonal factors [12]. The growth period with the greatest susceptibility to establishing future health conditions in adulthood is represented by the first thousand days of life [13]. The main regulators of growth are nutrition and hormonal factors, which are closely correlated and interdependent. However, it has been observed that nutrition plays a predominant role during the first two years of life, whereas hormonal factors become the main regulators of growth after the second year of life [14]. The growth hormone regulatory system includes several molecules such as growth hormone (GH), insulin-like growth factor 1 (IGF-1), insulin, thyroid hormones, sex hormones, and leptin. The GH/IGF-1 axis plays a central role in growth [15]. GH is synthesized by the somatotropic cells of the adenohypophysis under the stimulus of hypothalamic GH-releasing hormone (GHRH) and carries out its action both directly by binding to the GH receptor (GHR) expressed in many tissues, particularly in growth cartilage, and indirectly through the action of IGF-1, which is synthesized in the liver [16]. Insulin is an important anabolic hormone that contributes to growth during fetal life. The structure of insulin shows a high homology with IGF-1, exerting its action by binding to its receptors [17]. Other hormones implicated in postnatal growth are thyroid hormones and, during the pubertal development period, sex hormones such as estrogen and testosterone [18]. Finally, it has been observed that leptin, a hormone synthesized by white adipose tissue cells, is also implicated in the regulation of GH synthesis through stimulation of hypothalamic neurons [14]. In turn, hormonal factors are influenced by nutritional status. It has been shown that in cases of malnutrition, both IGF-1 levels and GHR expression decrease; in addition, insulin and leptin synthesis are reduced in cases of inadequate nutrition. It has also been observed that the expression of fibroblast growth factor 21 (FGF21) increases, suppressing both the GH/IGF-1 axis and chondrogenesis in the growth cartilage [14]. While malnutrition correlates with poor growth, excess weight during childhood is associated with taller stature, early pubertal development with accelerated bone age, and reduced growth velocity during the pubertal growth spurt. This effect is correlated with increased levels of leptin, insulin, IGF-1, and sex hormones; in contrast, GH levels in children with obesity are decreased, demonstrating that growth gain in obese children is not dependent on GH action [19]. Intrauterine development represents the first stage of growth and the one most involved in determining the future phenotypic characteristics of the child. The main factors responsible for fetal development are genetic potential and maternal and placental factors [20]. Fetal growth can be assessed both during intrauterine development through ultrasound and Doppler flowmetry and at birth through direct detection of auxological parameters. Newborns can be defined as small for gestational age (SGA), adequate for gestational age (AGA), or large for gestational age (LGA), with a weight and/or length below the 10th percentile, between the 10th and 90th percentiles, and above the 90th percentile, respectively, relative to the reference population [21]. The concepts of SGA and IUGR are often considered equivalent, but it must be remembered that they define different situations, although the two conditions may coexist. IUGR is defined as a fetus that fails to reach the estimated growth potential for its gestational age, and this condition is diagnosed early during intrauterine development with at least two findings on fetal ultrasound of growth slowing or stunting. The main difference between SGA and IUGR is that SGA presents a linear and constant growth pattern throughout pregnancy, even if at the lower percentiles, whereas IUGR presents a growth curve characterized by deflection [22]. Exposure to unfavorable intrauterine conditions, such as malnutrition, leads to adaptive mechanisms that cause alterations in fetal growth and modifications in metabolism through epigenetic phenomena (Figure 1). According to Barker’s theory of fetal programming, humans early in life are plastic and adaptive, so one genotype can give rise to different phenotypes in response to different environmental stimuli. It has been observed that children born SGA and IUGR have a higher risk of developing alterations in glucose metabolism, IR, T2D, obesity, hypercholesterolemia, hypertension, and cardiovascular disease in adulthood [23]. Nutrition is a major regulator of growth and represents a crucial element in determining metabolic risk factors later in life [24]. The following chapters will describe the effects of malnutrition and/or inadequate nutrition during the various stages of child growth and their impact on health outcomes during adulthood (Table 1). 

## 4. Intrauterine Growth and Influence of Nutrition During Pregnancy

In recent years, many studies have been conducted in order to identify possible causes for early onset obesity in children. Obesity is an important risk factor for developing complications in adulthood, such as IR, T2D, hypertension, and increased cardiovascular risk [42]. It is well known that obesity is a multifactorial condition, and its development depends not only on genetics and lifestyle but also on factors acting during the intrauterine and perinatal periods [43]. Particularly, the strong impact of nutrition during the first 1000 days of life, which are crucial to metabolic programming and in determining the risk of developing metabolic diseases later in life, has been highlighted. Regarding intrauterine development, it has been observed that exposure to unfavorable environmental conditions, such as inadequate nutrition, causes heritable epigenetic changes that contribute to a predisposition for developing obesity and metabolic alterations [44]. Epigenetics is a branch of genetics that includes many phenomena resulting in heritable changes in gene expression without altering the DNA sequence. These mechanisms cause phenotypic changes in the absence of genotypic changes [45]. DNA is a nucleic acid consisting of two paired polynucleotide chains forming a double helix. It is associated with basic proteins called histones, and together they form chromatin. The degree of chromatin compaction determines accessibility to transcription factors and enzymes responsible for transcription and gene expression [46]. Some of the main epigenetic mechanisms are DNA methylation, histone modifications, genetic imprinting, and the action of noncoding RNAs. DNA methylation involves the transfer of a methyl group by the enzyme DNA methyltransferase at the cytosine in promoter regions, resulting in reduced expression of the related gene. The phenomenon of genetic imprinting also depends on DNA methylation, which regulates gene expression of parental alleles [47]. Regarding histone modifications, acetylation of histones results in reduced chromatin compaction, favoring gene transcription, whereas deacetylation has the opposite effect [48]. It has been observed that the period of embryonic implantation is one of the most susceptible times for the development of epigenetic modifications, although these may arise throughout intrauterine life depending on exposure to unfavorable environmental factors [49]. One of the most significant factors is maternal nutrition. An unbalanced intake of macronutrients and micronutrients, together with increased maternal pre-pregnancy BMI and excessive weight gain during pregnancy, is associated with poor metabolic outcomes for the child. Similarly, reduced BMI and insufficient weight gain during pregnancy are associated with poor fetal growth and unfavorable outcomes [50]. The association between fetal exposure to certain dietary patterns and epigenetic changes causing metabolic diseases has been widely highlighted in the literature.

### 4.1. Carbohydrates

Carbohydrates represent a major source of energy. They are differentiated into simple and complex carbohydrates according to their structure. Simple carbohydrates include monosaccharides, disaccharides, and oligosaccharides, while complex carbohydrates are represented by polysaccharides [51]. Carbohydrates can also be classified according to their glycemic index (GI), a parameter indicating how quickly blood sugar rises after a meal. Foods with a low GI include whole grains, legumes, fruits, vegetables, and dairy products, while those with a high GI include refined grains, sweets, and sugar-sweetened beverages [52,53]. Another important parameter to consider is the glycemic load (GL), which is determined by the GI and the amount of carbohydrates contained in a food [54]. A diet during pregnancy that favors foods with a low GI and is rich in fiber has many benefits for both mother and fetus. Conversely, the consumption of foods with a high GI increases the risk of GDM, preeclampsia, and impaired fetal development [55]. GDM is a major complication during pregnancy and is associated with unfavorable long-term outcomes for the offspring. A maternal hyperglycemic state stimulates fetal pancreatic β-cells to increase insulin synthesis. Insulin is one of the hormones implicated in fetal growth; consequently, fetal hyperinsulinemia results in the growth of fetal tissues, often leading to macrosomia at birth. Furthermore, fetal exposure to elevated blood glucose levels is associated with increased adipose tissue deposition during infancy, reduced insulin sensitivity, and altered functional integrity of β-pancreatic cells [56,57,58]. This is likely due to epigenetic modifications in response to the hyperglycemic state, such as DNA methylation involving genes like hepatocyte nuclear factor alpha (HNF4α) and Ras-responsive element-binding protein 1 (RREB1) [59]. It has also been observed that restricted carbohydrate consumption during pregnancy is associated with poor fetal growth outcomes. Reduced carbohydrate intake is linked to defects in neural tube development and an increase in fat and protein consumption, leading to epigenetic changes that favor weight gain during infancy and the development of metabolic disorders [50]. Therefore, a diet characterized by adequate carbohydrate intake with a preference for low-GI foods and fiber appears to be associated with better glycemic control and greater insulin sensitivity in the mother, resulting in better fetal development outcomes and reduced metabolic complications in adulthood [25,26,27,28].

### 4.2. Proteins

The diet of healthy women in the preconception period and throughout most of pregnancy has a significant effect on birthweight, and proteins are the macronutrients with the greatest influence [29]. It was observed that both high and low protein intake may reduce fetal growth; hence, current recommendations suggest a maternal dietary intake of 10–20% of the total energy intake which is correlated with an increased birthweight and decreased risk of SGA [30,31]. Maternal protein deficiency is a pattern of diet also associated with an increased risk of developing alterations in glucose metabolism. Several genes have been identified through animal models whose expression changed due to epigenetic modifications, primarily represented by DNA methylation [60]. It has been observed that low-protein dietary patterns are associated with increased methylation of the HNF4α gene, resulting in reduced expression. This gene is implicated in pancreatic cell development and the control of insulin synthesis; its decreased expression contributes to the early development of T2D [61]. Another example is the reduced methylation of the IGF-2 promoter in the offspring of mothers exposed to malnutrition during pregnancy. This gene is involved in fetal growth, and alterations in its expression lead to the development of metabolic syndrome [62]. In a mouse model, changes in the promoter methylation of the leptin gene were also observed; leptin is a hormone synthesized by white adipose tissue cells that regulates appetite. Consequently, alterations in its expression and impaired levels of the hormone could increase the risk of obesity [63]. In another study conducted in a pig model, reduced methylation of the gene encoding for the enzyme glucose-6-phosphatase (G6PC) was demonstrated, resulting in increased expression and a consequent hyperglycemic effect [64]. Other examples include reduced methylation of the peroxisome proliferator-activated receptor alpha (PPARα) and glucocorticoid receptor (GR) genes; increased expression of these genes is associated with an increased risk of metabolic syndrome and cardiovascular disease [65]. Moreover, it has been highlighted that a maternal low-protein diet, especially during the last week of gestation, is associated with β-pancreatic cell dysfunction through increased expression of microRNA-143, resulting in a reduction in insulin synthesis and increased susceptibility to glucose intolerance in adulthood [66]. Adequate protein intake is essential to ensure the requirements of amino acids such as methionine, serine, and glycine, which are necessary not only for protein synthesis and fetal growth but also for 1-Carbon metabolism, which governs many epigenetic mechanisms and cell proliferation and differentiation [67]. Alterations in these processes cause epigenetic modifications, mainly represented by DNA methylation, which determine changes in the expression of many genes involved in glucose metabolism, increasing susceptibility to developing T2D in adulthood.

### 4.3. Fatty Acids

Maternal diets rich in saturated fats have been widely employed in animal models to study adverse offspring outcomes. Relatively little information currently exists concerning the impact of maternal high fat diet (HFD) on offspring health in humans. Maternal exposure to an HFD results in increased bodyweight at weaning and in adulthood along with increased adiposity, blood lipid levels, and hyperinsulinemia and with a preference for high-fat foods [32,33]. This occurs through epigenetic modifications, primarily represented by DNA methylation. Increased methylation of the adiponectin and leptin receptor genes and decreased methylation of the leptin gene have been observed. Altered expression of these genes, which code for hormones synthesized by white adipose tissue that regulate the satiety and appetite-sense system, promotes the development of metabolic syndrome [68]. Reduced methylation has been observed in the μ-opioid receptor (MOR) and preproencephalin genes, which regulate eating behavior, leading to increased food intake and consequently an increased risk of obesity [65]. Exposure to an HFD appears to be associated with reduced cognitive development, increased aggressive behaviors, and depressed mood [69]. Intrauterine exposure to an HFD increases the risk of the offspring developing hepatic steatosis through altered methylation patterns of genes implicated in intrahepatic lipid accumulation and liver fibrosis [70]. Moreover, the correlation between an HFD and hypertension has been highlighted and explained. An HFD contributes to the development of hypertension through several mechanisms, such as deficiency of endothelial nitric oxide synthesis, ineffectiveness of the renin-angiotensin-aldosterone system, and oxidative stress [71]. Unlike saturated fatty acids, polyunsaturated fatty acids are key molecules for fetal growth. Adequate intake of these substances results in a reduction in preterm births, low birth weight, and IUGR, thereby improving perinatal and maternal outcomes [34].

### 4.4. Micronutrients

Micronutrients are essential components for adequate fetal growth. A balanced and varied diet should ensure sufficient intake of all micronutrients; however, supplementation during pregnancy is always recommended for some, such as folic acid and iron [72]. B vitamins are water-soluble and crucial for many metabolic processes and the development of the central nervous system. Specifically, vitamin B12 (cyanocobalamin) and vitamin B9 (folic acid) act as coenzymes in 1-carbon metabolism, converting homocysteine to methionine and forming methyl groups necessary for DNA and RNA methylation and cell proliferation and differentiation. B vitamins are found in foods of animal origin, as well as in legumes and leafy vegetables [73]. Vitamin D and calcium are essential for maintaining the integrity of calcium–phosphorus metabolism and bone development. Vitamin D is a fat-soluble vitamin, and its deficiency during pregnancy is associated with risks of rickets in newborns, GDM, preeclampsia, and preterm births [74]. Iron is important for many enzymatic processes, gene regulation, and the synthesis of hemoglobin and myoglobin. Iron deficiency during pregnancy is associated with an increased risk of SGA, preterm births, and low birth weight infants [75]. Iodine is another key element essential for thyroid hormone synthesis. During pregnancy, the synthesis of thyroid hormones increases, and they are critical for fetal and central nervous system development. Therefore, a dietary iodine intake of 250 μg/day is recommended [76]. Adequate micronutrient intake is critical for proper fetal development and better maternal outcomes. Deficiencies in these substances can be avoided by adopting a balanced and varied diet; however, when this is not possible, supplementation is considered necessary.

### 4.5. Paternal Malnutrition

The impact of maternal diet during pregnancy on fetal development has been widely debated in the literature. However, recent studies have observed that paternal malnutrition also influences the metabolic characteristics of the offspring through heritable epigenetic changes, primarily represented by DNA methylation. Improper dietary patterns, such as a diet low in protein and high in fat, cause alterations in Sertoli cells and spermatogenesis [77]. Many genes have been identified in this context, such as PPARα, whose promoter is hypermethylated, and the IL-13 receptor α2 (Il13ra2) gene, which is hypomethylated, both of which are implicated in the development of T2D [65]. Furthermore, it has been observed that a high-fat paternal diet is an additional risk factor for the development of metabolic syndrome due to epigenetic changes involving the adiponectin and leptin genes [78]. These epigenetic changes contribute to an increased susceptibility of the offspring to develop metabolic disorders. It was also shown that a higher intake of junk food was associated with a higher risk of paternal sperm methylation at sites such as MEG3-IG, which are associated with obesity. Other epigenetic alterations such as DNA methylation or histone modification can be caused by the action of Reactive Oxygen Species (ROS) released from food with high carbs and high fats by altering testicular metabolism [79]. Noor et al. demonstrated the correlation between paternal obesity and persistent DNA methylation in the epigenome in offspring, suggesting that the DNA changes can be persistent [80].

## 5. Extrauterine Growth Restriction

Another particular condition associated with altered growth patterns and unfavorable cardiometabolic outcomes is defined extrauterine growth restriction (EUGR). EUGR is a condition primarily affecting premature infants (especially those born before 32 weeks), particularly those with very low birth weight (VLBW, <1500 g) or extremely low birth weight (ELBW, <1000 g), who miss the critical period of growth that occurs during the last trimester, which is essential for the accumulation of fat, muscle, and nutrients [81]. It occurs when an infant’s growth rate falls below the expected standard for their post-conceptional age, resulting in growth parameters (such as weight, length, or head circumference) that are significantly below normal levels for their age. EUGR is diagnosed when weight is <10th percentile at either discharge or 36–40 weeks postmenstrual age, although a consensus definition of EUGR is still lacking [82]. This condition is commonly observed in premature infants admitted to NICUs because their caloric and protein requirements are physiologically higher due to critical illnesses such as respiratory distress, pulmonary bronchodysplasia, sepsis, intraventricular hemorrhage, patent ductus arteriosus, and necrotizing enterocolitis; moreover, since their digestive systems are immature and can tolerate only limited nutrition, they must be supported with parenteral nutrition. Much evidence has found that prematurity and growth restriction before 40 weeks of correct gestational age (CGA) increased the risk of adverse neurodevelopmental and metabolic outcomes. Martìnez-Jimènez et al. [83] reviewed the evidence regarding EUGR-related comorbidities in childhood by a systematic approach including 19 cohort studies. It was observed that EUGR is mainly associated with poor growth and neurodevelopment in childhood and cardiometabolic alterations such higher systolic and diastolic blood pressure, lower HDL-C, and higher glycemic plasma levels. From the reported data, it is obvious that very premature neonates require close monitoring of their growth and adequate administration of nutrients, both via the parenteral and enteral route, especially in the first 2 weeks of life by universal common feeding protocols.

## 6. Infant Feeding

The breastfeeding period represents one of the crucial stages of growth. It is well known that there are differences in the short- and long-term effects on a child’s metabolism depending on whether breastfeeding or formula feeding is carried out. Although exclusive breastfeeding is strongly recommended for the first six months of life, this is not always possible due to various factors [84]. One study observed that mothers with a high pre-pregnancy BMI who developed GDM had greater difficulty initiating breastfeeding. An association was found between GDM and delayed initiation of lactogenesis [85]. The following chapters will describe the main properties of breast milk, the differences compared to formula milk, and the corresponding effects on children’s metabolism.

### 6.1. Breastfeeding

Breast milk is classically considered the best food for infants during the lactation period. The WHO recommends exclusive breastfeeding for the first six months of life and emphasizes its beneficial effects up to 12 months when combined with other complementary foods [86]. Breastfeeding has numerous benefits for both the baby and the mother. The composition of breast milk varies greatly among individuals, influenced by genetic factors, maternal health, pregnancy duration, lifestyle, and diet during pregnancy and lactation (Figure 2). The composition of breast milk also changes over the course of lactation. Colostrum is produced in the first few days after delivery, is rich in serum proteins, such as secretory immunoglobulins (IgA), and has a lower concentration of fat and lactose. It contains high levels of growth factors like epidermal growth factor, transforming growth factor-beta, and colony-stimulating factor, serving primarily an immunological function. Over the next few days, breast milk transitions to mature milk, which has higher concentrations of fat and lactose to support nutritional needs and promote growth. Pre-term milk tends to have a higher concentration of serum proteins compared to full-term milk [87]. Breast milk proteins have nourishing, antimicrobial, and immunomodulatory functions. They are classified into serum proteins, caseins, and mucins. Serum proteins, including alpha-lactalbumin, lactoferrin, lysozyme, and IgA, are rich in essential amino acids. Caseins, synthesized in the mammary gland, are insoluble proteins with high levels of methionine and proline. Mucins are membrane proteins associated with lipid droplets in milk. Digestive enzymes like alpha1-antitrypsin, lipase, and amylase are also present [88,89]. The quality and quantity of breast milk proteins significantly influence infant body composition and growth. Higher protein content in milk has been linked to weight gain and increased obesity risk due to the activation of the insulin-like growth factor axis [90]. Research indicates that the protein composition of breast milk can vary significantly between obese and normal-weight women. One of the notable differences observed is the increased concentration of branched-chain amino acids (BCAAs) in the breast milk of obese women. This variation may be linked to a heightened risk of obesity in children, although more research is needed to fully understand this relationship [91]. The carbohydrate component of breast milk includes lactose and HMOs. Lactose provides nutritional energy, while HMOs, which vary based on maternal genetics and enzyme expression in the mammary glands, function as prebiotics. They promote the growth of beneficial intestinal bacteria like Bifidobacteria and protect against pathogens like Salmonella, Listeria, and Campylobacter [92,93]. It has been observed that maternal overweight and obesity can affect the quantity and quality of HMOs, increasing the risk of childhood obesity. Specifically, low levels of fucosylated and sialylated HMOs, along with increased levels of 3-fucosyllactose, 3-sialyllactose, and 6-sialyllactose, have been identified in the breast milk of obese mothers. These changes in HMO composition may influence infant gut microbiota and metabolic programming, thereby contributing to a higher risk of obesity [94]. Furthermore, maternal obesity has been observed to be related to an increase in non-glucose monosaccharides such as mannose in breast milk, which appear to be related to an increased risk of weight gain in the child after the first six months of life [3]. Lipids, mainly triglycerides, are the primary energy source in breast milk. They are secreted as lipid droplets with a nonpolar core of triglycerides and an outer membrane containing glycerolphospholipids, sphingolipids, sphingomyelin, and glycosylated proteins. These substances are crucial for central nervous system development and myelination [95]. It has also been observed that lipids in breast milk, such as medium-chain monoglycerides, may inactivate many pathogens such as group B streptococcus. Maternal diet and BMI influence the type and concentration of fats in breast milk, with higher saturated fat levels observed in overweight or obese women [96]. Breast milk also contains essential vitamins and minerals. The vitamin content depends on the maternal diet. For instance, vitamin B12 deficiency can occur in infants of mothers following vegetarian or vegan diets, posing a risk for neurodevelopmental issues. Vitamin K, which is low in breast milk, is administered at birth to prevent hemorrhagic disease in newborns. Vitamin D, also deficient in breast milk, should be supplemented during the first year of life. Other important vitamins include vitamins C, A, and E [97]. Breast milk contains microRNAs, which regulate gene expression and cell development [98]. Altered miRNA content has been documented in women with overweight/obesity; specifically, miRNA 30b and miRNA 30c appear to be associated with adipogenesis and altered glucose metabolism in offspring [99,100]. Moreover, hormones like leptin, adiponectin, and insulin also contribute to metabolic development. Additionally, maternal intestinal microbiota bacteria are present in breast milk, allowing it to function both as a prebiotic and probiotic [101]. It is well known that exclusive breastfeeding is associated with numerous positive short- and long-term health effects. Short-term effects include decreases in respiratory and gastrointestinal tract infections, resulting in reduced hospitalization and mortality rates in early life. Long-term benefits include a reduced risk of developing overweight, obesity, and T2D in adulthood [35,36]. Indeed, it has been observed that breastfed infants tend to develop healthier eating habits, showing less food selectivity and a preference for fruits and vegetables when complementary foods are introduced, likely due to their exposure to a wider range of tastes [102].

### 6.2. Formula Feeding

Formula milk is designed to provide essential nutrients such as proteins, lipids, carbohydrates, minerals, and vitamins in concentrations established by nutritional guidelines. Typically based on cow’s milk due to its similarity to human milk, or alternatively soy milk, formula milk undergoes specific modifications to better match human milk. These modifications include reducing casein and saturated fat content, adjusting the calcium/phosphorus ratio, and adding iron. Despite these adjustments, significant differences in composition persist, potentially increasing the long-term risk of obesity, T2D, and cardiovascular disease in formula-fed infants [103]. Differences have been highlighted especially regarding fat and protein content. Formula milk generally contains a lower concentration of polyunsaturated fatty acids compared to breast milk; these fatty acids are crucial for brain development and function, and their lower levels in formula milk may impact neurodevelopmental outcomes [104]. In contrast, the protein content of infant formula remains higher than that of breast milk, leading to faster growth rates in infants. This increased growth is associated with higher plasma levels of IGF-1, which has been linked to a higher risk of obesity and IR later in life [105]. Moreover, formula milk lacks HMOs, resulting in differences in the gut microbiota of formula-fed infants compared to breastfed infants. The gut microbiota is critical for nutrient absorption, immune protection, and metabolic programming. Early alterations in the gut microbiota have been linked to an increased risk of overweight and obesity during childhood [37].

## 7. Weaning and Complementary Feeding

Complementary feeding begins after six months of age, marking a critical transition period in which solid foods are introduced while breast or formula milk consumption is gradually reduced. This phase, extending until around two years of age, is essential for ensuring adequate nutrient intake as the infant’s nutritional needs evolve [106]. During this period, cow’s milk should not be introduced before 12 months due to its potential to cause iron deficiency and provide excessive protein [107]. By six months, infants often experience iron deficiency due to increased utilization for blood volume expansion and neurocognitive development [108]. The recommended daily iron intake for infants ages 6–12 months ranges from 0.9 to 1.3 mg/kg/day. To meet this requirement, iron-enriched foods, such as iron-fortified cereals, are particularly recommended for breastfed infants. Upon the initiation of complementary feeding, protein intake tends to rise, often exceeding recommended levels. Protein should account for 8–12% of total daily caloric intake. Many studies have suggested that an excessive protein consumption during early childhood has been linked to increased weight gain, higher BMI, and greater body fat percentage later in life [39]. This association is explained by the “early protein hypothesis”, which posits that high protein intake stimulates the IGF-1 axis and the mammalian target of rapamycin (mTOR) pathway, promoting growth and adipogenesis [106,109]. High protein intake results in elevated levels of essential amino acids, particularly BCAAs and serum urea nitrogen, which stimulate insulin and IGF-1 production. BCAAs, especially leucine, can reduce fatty acid oxidation and increase insulin release, while aromatic amino acids further stimulate IGF-1 secretion [109]. These mechanisms contribute to rapid growth and increased adiposity, reinforcing the importance of monitoring protein intake during complementary feeding. The timing of complementary feeding introduction significantly influences BMI and the risk of developing overweight [38]. In a meta-analysis of 13 prospective cohort studies, Wang et al. highlighted that early introduction of complementary foods (before 4 months) in infants breastfed for less than 4 months or formula fed was associated with a higher risk of overweight during childhood [38]. Early introduction also poses risks for food allergies, celiac disease autoimmunity due to gluten-containing foods, and T1D [110]. Breastfeeding during the introduction of solid foods offers protective benefits against inflammatory diseases and chronic conditions. It mitigates oxidative damage to the gut microbiota and supports gut maturity, reducing the risk of food allergies [111]. Additionally, breastfeeding has been shown to provide protective factors against oxidative stress during complementary feeding.

## 8. Nutrition in Childhood

Obesity is a significant public health issue linked to numerous metabolic and cardiovascular risks. Despite its prevalence, it often goes unnoticed, as it is sometimes perceived as a healthy status by public opinion [112]. Nutrition plays a critical role in obesity development among children and adolescents, influencing gut microbiota composition, which is integral to the “gut–brain axis”. A diet high in animal proteins, saturated fats, and sugars can lead to the loss of beneficial gut bacteria, whereas consuming complex polysaccharides and plant proteins promotes the production of short-chain fatty acids (SCFAs) like acetate, propionate, and butyrate. SCFAs reduce intestinal inflammation markers and stimulate the release of peptide YY and Glucagon-like peptide 1 (GLP-1), both of which have anti-obesity and anti-diabetic effects [113,114] Rampelli et al. found that diets high in fat and carbohydrates or proteins were associated with an increased risk of obesity and a low diversity gut microbiota, rich in obesogenic bacteria like Prevotella, Dorea, and Bacteroides. The interaction between individual gut microbiota and diet type can predict obesity risk in children [115]. The Mediterranean diet, characterized by vegetables, nuts, whole grains, legumes, moderate fish intake, and reduced consumption of poultry and dairy, with olive oil as the main fat source, is beneficial [116]. Adherence to this diet can be assessed using indexes like the Mediterranean Diet Quality Index for children and adolescents (KIDMED) [117]. Early adherence to the Mediterranean diet can prevent obesity [118,119] and reduce the risk of T2D through interactions with the melanocortin-4 receptor gene [118,120]. The diet’s anti-inflammatory effects and components like olive oil reduce IR, as evidenced by lower CRP and adiponectin levels in diabetic patients. Moreover, antioxidants in fruits, vegetables, and cereals reduce oxidative damage to pancreatic β-cells [121]. The HELENA (Healthy Lifestyle in Europe by Nutrition in Adolescence) study reported that the Mediterranean diet reduced waist circumference, an indicator of abdominal adiposity, in obese adolescents with genetic risk [118]. Similarly, a previous study reported that adherence to this diet for 12 months improved triglycerides and LDL-C levels, correlating with better carotid intima-media thickness values in pre-pubertal hypercholesterolemic children [122]. The consumption of sugar-sweetened beverages (SSBs) among children has increased, exceeding recommendations. Despite mixed results from previous reviews, a systematic review by Rusham et al. confirmed a positive association between SSBs and increased BMI, percentage body fat, and overweight/obesity [40]. SSBs contain added caloric sweeteners like sucrose, high fructose corn syrup, and fruit juice concentrates. The UK Scientific Advisory Committee for Nutrition (SACN) recommends limiting free sugar intake to less than 5% of total daily energy for children older than 2 years [123]. A systematic review demonstrated that interventions in preschool settings have successfully reduced SSB consumption in children ages up to 5 years [124]. Skipping breakfast is common among adolescents in Western countries. The HELENA study, which included adolescents from nine European countries, found that regular breakfast consumption was associated with lower body fat [125]. A good quality breakfast, often including cereals, correlates with lower BMI, and regular breakfast consumption positively affects weight and cardiometabolic profiles, with lower waist circumference, triglycerides, LDL-C, TC/HDL-C, and LDL-C/HDL-C ratios [125]. In a systematic review and meta-analysis Wang et al. evidenced that skipping breakfast is associated with an increased risk of overweight/obesity in children and adolescents [41]. Similarly, Jeans et al. suggested how breakfast consumption was effective in improving glycemia control in high-risk pediatric populations [126]. In children, obesity often correlates with non-alcoholic fatty liver disease (NAFLD), characterized by at least 5% macrovesicular or microvesicular steatosis. Risk factors include high fat and carbohydrate intake and a sedentary lifestyle. High fructose intake is linked to increased lipogenesis in the liver and decreased hepatic lipid oxidation due to PPAR-α inhibition, inducing oxidative stress via cytochrome P450-2E1 activation. Although the Mediterranean diet reduces IR and plasma inflammatory biomarkers in children with NAFLD, there is no consensus on the best diet for treating NAFLD [127]. Studies indicate lower concentrations of omega-3 polyunsaturated fatty acids (n-3 PUFA) and a high n-6/n-3 PUFA ratio in NAFLD patients, recommending n-3 PUFA supplementation to improve liver steatosis [128,129]. Moreover, supplementing docosahexaenoic acid (DHA) and vitamin D in vitamin D-deficient children with NAFLD improves IR, transaminases, and triglyceride levels [130].

## 9. Conclusions

Growth represents a crucial and complex developmental stage influenced by multiple factors, with diet playing a pivotal role in determining metabolic alterations. These alterations, including obesity and insulin resistance, significantly contribute to cardiovascular risk. To mitigate the onset of cardiovascular diseases in adulthood, early lifestyle interventions are essential. Fundamental is parental nutrition education, promoting a healthy and balanced diet that includes all macronutrients and micronutrients. Indeed, the health of the future unborn child is nothing but a consequence of the lifestyle of the ancestors. Preventing gestational diabetes mellitus during pregnancy, promotion of breastfeeding during infancy, and having healthy dietary and lifestyle habits during childhood might be helpful strategies. School programs should promote nutrition awareness, education campaigns offering healthy snacks and balanced lunches, and increasing school hours of physical education. At-home parents should encourage the reduction of time spent in front of screens and sedentary behaviors, promoting outdoor activities and games. The acquisition of correct eating habits and a healthy lifestyle from an early age might be the key to reducing the incidence of cardiovascular diseases, highlighting the importance of early prevention and health promotion.

## Figures and Tables

**Figure 1 nutrients-16-03801-f001:**
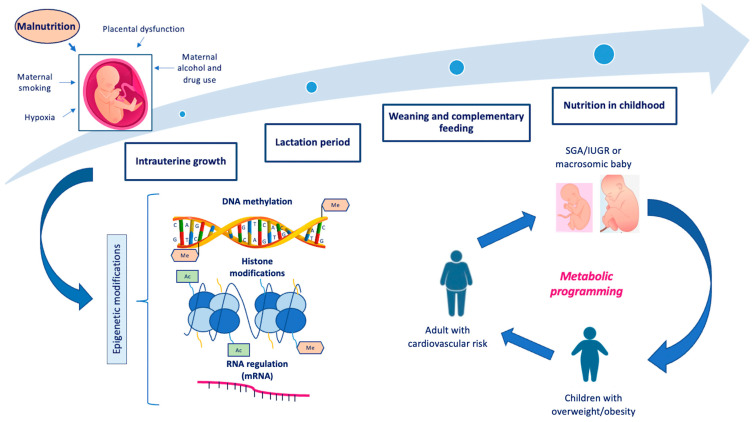
Nutrition during growth and its influence on metabolic programming and developing metabolic diseases later in life. Abbreviations: SGA: small for gestational age; IUGR: intrauterine growth restriction.

**Figure 2 nutrients-16-03801-f002:**
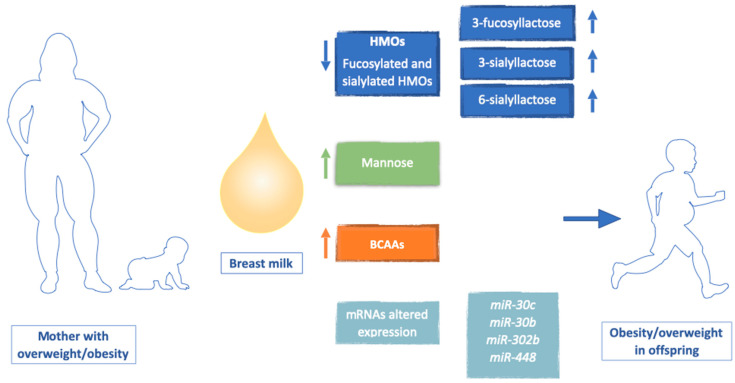
Altered breast milk composition in overweight/obese mothers represents a risk factor for increased BMI in offspring. Abbreviations: HMOs: human milk oligosaccharides; BCAAs: branched-chain amino acids.

**Table 1 nutrients-16-03801-t001:** Summary of the selected studies on the effect of nutrition at different growth stages on the risk of developing obesity and metabolic syndrome later in life.

	Author (Year), *n*	Study Design	Main Findings
Intrauterine Growth			
Carbohydrates	Geiker et al. (2022), *n* = 279, [25]	RCT	A moderate increase in dietary protein in conjunction with a reduction in glycemic index during the last two trimesters of pregnancy, reduced gestational weight gain, and limited complications and cesarean deliveries among women with overweight or obesity.
	Wei et al. (2016), *n* = 302, [26]	Systematic review and meta-analysis of five RCTs	A low-GI diet is associated with a significantly lower risk of macrosomia and insulin usage in women with GDM.
	Viana et al. (2014) *n* = 257, [27]	Systematic review and meta-analysis of four RCTs	A low-GI diet was the only confirmed advantageous dietary intervention in patients with GDM and was associated with less frequent maternal insulin usage and lower newborn birth weight.
	Zhang et al. (2016), *n* = 1985, [28]	Systematic review and meta-analysis of 11 RCTs	Low-GI diets significantly reduced fasting blood glucose in healthy pregnant women, women with gestational hyperglycemia, and women with pregestational T2D and the proportion of LGA.
Proteins	Cucò et al. (2010), *n* = 77, [29]	Longitudinal study	Diet of healthy women in the preconception period and throughout pregnancy has a significant effect on birth weight, and proteins are the macronutrient with the greatest influence.
	Ota et al. (2015), *n* = 6705, [30]	Systematic review of 12 RCTs	Balanced energy and protein supplementation (<25% energy from protein) is associated with increased birthweight and a decreased risk of stillbirth and SGA.
	Morisaki et al. (2018), *n* = 91,637, [31]	Prospective cohort study	An inverse U-shaped curve describes the relationship between the risk of fetal growth restriction and protein intake, suggesting that both high and low protein intake can increase its risk. An increase in protein density up to 12% was associated with increased fetal growth and reduced SGA, and any further increase in protein density significantly reduced fetal growth.
Fatty acids	Chaves et al. (2021), [32]	Systematic review of 17 animal studies	Maternal exposure to a HFD led to increased food intake and increased preference for HFDs. The offspring from HFD mothers presented with low birthweight but became heavier into adulthood. In addition, these animals also exhibited greater fat deposition on white adipose tissue pads.
	Ribaroff et al. (2017), *n* = 6047 offspring, [33]	Systematic review and meta-analysis of 171 animal studies	Maternal exposure to HFD results in increased bodyweight at weaning and in adulthood along with increased adiposity, blood lipid levels, and hyperinsulinemia in both male and female offspring.
	Middleton et al. (2018) *n* = 19,927, [34]	Systematic review and meta-analysis of 70 RCTs	n (omega)-3 PUFA supplementation was associated with reduced risk of preterm birth and low birthweight.
Infant feeding	Horta et al. (2019), *n* = 250,000, [35]	Systematic review and meta-analysis of 14 studies	Breastfeeding has a benefit on protecting against T2D, and this protection seems to be greater among adolescents, suggesting that the benefit of breastfeeding may decrease overtime.
	Sun et al. (2024), *n* = 2769, [36]	Cross-sectional study including participants ages 2–6 years old	Children who breastfed for a longer period of time were less likely to be overweight or obese at the age of 2 to 6.
	Forbes et al. (2018), *n* = 1087, [37]	Multicenter study	Breastfeeding is protective against overweight, and the gut microbiota contribute to this effect. Formula feeding was associated with higher microbiota diversity, and this partially explained the increased risk of overweight among non-breastfed infants.
Weaning	Wang et al. (2016), *n* = 63,605, [38]	Meta-analysis of 13 prospective cohort studies	Introducing complementary foods before 4 months of age compared to at 4 to 6 months was associated with an increased risk of being overweight or obese during childhood.
	Pearce et al. (2013), *n* = 4486, [39]	Systematic review of 10 studies	High intakes of energy and protein, particularly dairy protein, in infancy could be associated with an increase in BMI and body fatness.
Nutrition in childhood	Rousham et al. (2022) [40]	Systematic review and meta-analysis of 60 studies	In children ≤10.9 years, consumption of SSBs and unhealthy foods may increase BMI, body fat percentage, or odds of overweight/obesity.
	Wang et al. (2023), *n* = 323,244, [41]	Systematic review and meta-analysis of 40 studies	Skipping breakfast is associated with an increased risk of overweight/obesity in children and adolescents.

## Data Availability

All data are contained within the article.

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
