# Peer review of "Influence of Nutrition on Growth and Development of Metabolic Syndrome in Children"

_nutrients, 2024, doi:10.3390/nu16223801_

Round 1

Reviewer 1 Report

Comments and Suggestions for Authors

The research on the title "Influence of Nutrition on Growth and Development of Metabolic Syndrome in Children" by the authors Quarta and their collaborators was very interesting to examine. I would like to congratulate them, but I believe they should work on improving the following.

The abstract, in my opinion, needs to be revised and improved. It should not only restate the primary subject matter in a large number of lines, but it should also include the search process and what families ought to do, in addition to the practical applications of the results, which are also not very clear.

METHODS: Is there simply no methodology to be found? How did the information come to be collected?

We are aware that it is a review of the literature; however, would it not be possible to create a systematic approach to it and incorporate a PRISMA flow? The topic is very narrowly focused, and I believe that the studies ought to be highlighted in a more concrete manner, even if it means adding a table with authors who are also relevant.

Overall, there are references that, in my opinion, ought to be brought up to date; references that date back to 2001 indicate that 23 years have passed.

Personally, I believe that point 3.5 could be improved by providing examples of parental diets and the considerations that go into them.

Additionally, there is a need for a section on practical applications; the next steps are not entirely clear.

It is recommended that the conclusions be revised; however, we are not aware of the particular outcomes of this review search.

Taking into consideration all of these factors, I believe that the review article has a great deal of room for improvement; it offers general data, much like a chapter in a book, but it does not really offer any concrete applications.

Author Response

We thank the reviewers for their proofreading work and for all the suggestions offered. All the insights have been useful and provided us practical advice to improve the quality of the manuscript. We hope our work has met your expectations and trust that you’ll find it satisfactory. However, we remain at your disposal for any further improvements.

Comment 1: The abstract, in my opinion, needs to be revised and improved. It should not only restate the primary subject matter in a large number of lines, but it should also include the search process and what families ought to do, in addition to the practical applications of the results, which are also not very clear. 

Response: The abstract has been revised. We offered a comprehensive route of the topics examined in the text focusing on the major risk factors associated with unfavorable cardiometabolic outcomes. Methodology is treated in the appropriate section, and practical applications are discussed extensively in the respective sections of the text and summarized in the conclusions.

Comment 2: METHODS: Is there simply no methodology to be found? How did the information come to be collected? We are aware that it is a review of the literature; however, would it not be possible to create a systematic approach to it and incorporate a PRISMA flow? The topic is very narrowly focused, and I believe that the studies ought to be highlighted in a more concrete manner, even if it means adding a table with authors who are also relevant. 

Response: Our manuscript is a narrative review of the literature, providing an overview of the latest scientific evidence on nutrition and the related risk factors in the development of pediatric obesity and metabolic syndrome. Our aim is to propose strategies for prevention and intervention. By definition, the methodological approach of a narrative review does not involve adherence to the rigorous guidelines required for systematic reviews and meta-analyses, and thus, a PRISMA flow is not applicable to our work. However, we have included a section on materials and methods (lines 67-77) in the manuscript, and as suggested, we developed a table to highlight and summarize the main relevant studies.

Comment 3Overall, there are references that, in my opinion, ought to be brought up to date; references that date back to 2001 indicate that 23 years have passed. 

Response: We reviewed the entire bibliography and updated the references by removing those dated to before 2010.

Comment 4Personally, I believe that point 3.5 could be improved by providing examples of parental diets and the considerations that go into them. 

Response: We reviewed section 3.5 (corresponding to 4.5 in the revised text) adding more information on the epigenetics of paternal diet and some examples of dietary patterns related to unfavorable outcomes of metabolic programming.

Comment 5Additionally, there is a need for a section on practical applications; the next steps are not entirely clear. 

Response: Practical applications are included in each section of the text separately. For example, in the section on nutrition in childhood, the practical applications and prevention strategies mentioned are avoiding unhealthy eating habits such as skipping breakfast and consumption of sugar-sweetened beverages and promoting adherence to a Mediterranean diet. In any case a summary of all possible strategies and interventions at different growth stages can be found in the conclusion section.

Comment 6: It is recommended that the conclusions be revised; however, we are not aware of the particular outcomes of this review search.

Response: We revised the conclusion section focusing on preventive and intervention strategies.

Reviewer 2 Report

Comments and Suggestions for Authors

The article covers wide aspects of risk factors for childhood obesity, mostly concentrating on variety of food consumption at different age periods. The main question to me how the literature was selected, according which criteria. Here is narrative review, but some cleariness would be appreciated. No methods how was it done. 

Who is the author of the pictures? 

Some strange literature "break": from line 604 at literature numbering "58", it again starts from "1". Please, correct. 

Author Response

We thank the reviewers for their proofreading work and for all the suggestions offered. All the insights have been useful and provided us practical advice to improve the quality of the manuscript. We hope our work has met your expectations and trust that you’ll find it satisfactory. However, we remain at your disposal for any further improvements.

Comment 1: The article covers wide aspects of risk factors for childhood obesity, mostly concentrating on variety of food consumption at different age periods. The main question to me how the literature was selected, according which criteria. Here is narrative review, but some cleariness would be appreciated. No methods how was it done. 

Response: A paragraph on materials and methods has been added and highlighted in the main text and corresponds to the lines 67-77.

Comment 2: Who is the author of the pictures? 

Response: The pictures were created by first authors Alessia Quarta and Maria Teresa Quarta.

Comment 3Some strange literature "break": from line 604 at literature numbering "58", it again starts from "1". Please, correct. 

Response: There was an error in the insertion of the bibliography. It has been corrected.

Round 2

Reviewer 1 Report

Comments and Suggestions for Authors

The authors have improved their manuscript proposal. However, I consider that they can follow some more advice that they have decided not to follow such as separating the point from practical applications and changing the tables to the journal format. The first may be understandable but the second I consider that they should do it since the gray color in the letters is not commonly used.

Author Response

Comment: Thank you for the suggestions. We have changed the format of the table.